# The Prognostic Utility of Venous Blood Gas Analyses at Presentation in Cats with Cardiogenic Pulmonary Edema

**DOI:** 10.3390/vetsci10030232

**Published:** 2023-03-19

**Authors:** Akiyoshi Tani, Ryohei Suzuki, Satoshi Matsukata, Atsushi Nakamura, Takaomi Nuruki

**Affiliations:** 1The TRVA Animal Medical Center, Tokyo Jonan Regional Veterinary Medicine Promotional Association, Tokyo 158-0081, Japan; 2Laboratory of Veterinary Internal Medicine, School of Veterinary Medicine, Faculty of Veterinary Science, Nippon Veterinary and Life Science University, Tokyo 180-8602, Japan

**Keywords:** cat, cardiogenic pulmonary edema, blood gas, hypercapnia

## Abstract

**Simple Summary:**

Cardiogenic pulmonary edema (CPE) is the leading cause of cats visiting emergency hospitals with respiratory distress. The association of physical examination and venous blood gas parameters with the survival of cats with CPE was retrospectively investigated. Thirty-six cats were ultimately included, and eight of them died within 12 h after their presentation to our hospital. The Mann–Whitney U test with Bonferroni correction was used to compare the parameters. Cats that died within 12 h had significantly lower rectal temperatures and higher PvCO_2_ than those that did not die within 12 h. Moreover, vasoconstrictor use was related to death within 12 h of presentation and higher PvCO_2_. A large number of prospective studies should be performed to validate these results.

**Abstract:**

Cats urgently visit emergency hospitals due to respiratory distress, and the chief cause is cardiogenic pulmonary edema (CPE). Although cats with CPE were frequently encountered in clinics, the prognostic factors were poorly reported. The objective of this retrospective study was to investigate the association of physical examination and venous blood gas parameters with the survival of cats with CPE in an emergency hospital. Thirty-six cats with CPE were ultimately included in the present study, and eight of them died within 12 h after their presentation to our hospital. Statistical analyses of clinical parameters between cats that died within 12 h and those that survived for 12 h were conducted using Mann–Whitney U test with Bonferroni correction. Cats that died within 12 h had significantly lower rectal temperatures and higher PvCO_2_ than those that did not die within 12 h. Moreover, hypotension and vasoconstrictor use were related to death within 12 h of presentation and higher PvCO_2_. These findings indicated the prognostic utility of body temperature and PvCO_2_, and the association between hypercapnia and the severity of CPE or hypotension. A large number of prospective studies should be performed to validate these results.

## 1. Introduction

Congestive heart failure (CHF) is a life-threatening condition and a leading cause of emergency hospital visits for cats with respiratory distress [1]. Cardiomyopathies, including hypertrophic cardiomyopathy (HCM) and restrictive cardiomyopathy (RCM), are common heart diseases in cats [2]. Additionally, HCM has been diagnosed in approximately 15% of clinically healthy cats [2], while some cats develop acute congestive heart failure (CHF) resulting in cardiogenic pulmonary edema (CPE), pleural effusion, or occasionally arterial thromboembolism (ATE) [3]. Although respiratory distress due to pleural effusion is rapidly improved by thoracentesis, the symptom due to CPE usually needs more intensive care and hospitalization. In one study, the successful discharge rate of cats diagnosed with CHF who were urgently admitted to the hospital was 76% (42 of 55 cats) [4]. Nevertheless the low discharge rate, the prognostic factor of CHF have not been poorly reported without hypothermia [4].

The use of blood gas analysis in veterinary medicine has increased recently [5], and it can be conducted on arterial as well as venous samples. In comparison with venous blood gas analysis, arterial blood gas analysis yields accurate evaluation of oxygenation and ventilation but requires more advanced technical skills and involves stressful handling for cats. In contrast, while a venous blood sample is easier to obtain and venous blood gas analysis is useful to assess metabolic activity and ventilation status, it cannot accurately evaluate the ability of the lungs to oxygenate the blood and can be influenced by hemodynamic status. Venous blood gas analyses are useful tests to assess acid-base status measuring blood pH, PvCO_2_, HCO_3_^−^, and base excess (BE). The association between abnormal acid-base balance and mortality is reported in cats [6]. Venous blood gas analyses is, thus, used in conditions such as ATE, inflammatory lower airway disease, and chronic renal failure [7,8,9]. In feline ATE cases, hypercapnia is a negative prognostic factor, and a PvCO_2_ >34 mmHg taken from unaffected limbs is associated with an increased risk of death [8].

While venous blood gas analysis has been assessed for some feline diseases, it has not, to date, been evaluated in cats with CHF, which is a common cause of feline respiratory distress. While feline CHF cases are frequently represented with CPE and/or pleural effusion, CPE is not curable with thoracentesis and needs more intensive care. Furthermore, any prognostic factors, which includes blood gas analysis, were analyzed focusing on cats with CPE not CHF. Therefore, this study aimed to identify the signalment or venous blood gas predictive factors for mortality and to evaluate the association between PvCO_2_ and clinical data, including body temperature, heart rate, respiratory rate, and blood pressure, in cats diagnosed with cardiomyopathy-induced CPE at an emergency hospital.

## 2. Materials and Methods

### 2.1. Case Selection

Cats diagnosed with “CPE” at the TRVA Animal Medical Center between April 2019 and March 2021 were retrospectively included using a computerized search of the electronic medical records. The following exclusion criteria were used: (1) cats simultaneously diagnosed with ATE, (2) cats without blood gas analysis data within 1 h after their presentation, (3) cats treated using mechanical ventilation, and (4) cats without survival data after 12 h had passed after their presentation. The data from the first presentation were used if the cat had presented to the hospital multiple times with CPE during the included interval. At the hospital, when a dyspneic cat was admitted, immediate oxygen supplementation, physical exam, blood test, and thoracic radiographic and ultrasonographic tests were conducted, if acceptable for cats and their owners. As “physical exams”, body weight, body temperature, heart rate, and respiratory rate were measured in all patients. Non-invasive blood pressure or SpO_2_ was measured if possible. As a “blood test(s)”, blood gas analysis, CBC, and/or biochemistry tests were selected by each veterinarian in charge. As an “ultrasonographic test”, point of care ultrasonography [10,11] was conducted in all patients. Standard echocardiography or abdominal ultrasound was conducted if needed by the decision of each veterinarian in charge.

A retrospective evaluation of clinical data using all medical records and any imaging studies (thoracic radiography and ultrasonography) was conducted by two clinicians to validate the diagnosis of CPE. The presence of radiographic evidence of cardiomegaly and regional or diffuse interstitial or alveolar opacity was used to help the diagnosis of CPE, as previously reported by Ward et al. [11]. Because some cats showed signs of respiratory distress, the records of focused point of care ultrasonography [10,11] with or without echocardiography and abdominal ultrasound, which was performed by veterinarians experienced in emergency rescue, were retrospectively reviewed. The presence of pericardial or pleural effusion, and enlargement of the left atrial to aortic diameter were assessed as a diagnostic basis for determining cardiogenic causes. A cardiologist independently examined and reconfirmed these retrospective diagnostic processes blindly and carried out the final diagnosis of the phenotype of cardiomyopathy. Cats with left ventricular hypertrophy and absence of the other diseases known to cause left ventricular hypertrophy were diagnosed as HCM phenotype [3,12,13]. Cats with normal left ventricular dimension and wall thickness with left atrial or biatrial enlargement and a prominent endomyocardial scar that bridges the interventricular septum and left ventricular free wall were diagnosed as RCM phenotype [12,14]. Cats with reduced systolic function were diagnosed as dilated cardiomyopathy phenotype [3].

Treatment methods, drugs, the interval of hospitalization, and prognosis until discharge were retrospectively reviewed. Since the hospital is mainly for emergency patients during the night, we set 12 h as an interval to evaluate the short prognosis. Cats that died within 12 h were categorized as the “12-h death group” and those that survived beyond 12 h were categorized as the “12-h survival group”.

Among the clinical data on admission, body weight, body temperature, heart rate, respiratory rate, and blood pressure were recorded. Body temperatures below 32.0 °C were recorded as 32.0 °C since temperatures below this cannot be measured. Blood pressure was non-invasively measured using the oscillometric method. The data of jugular vein venous blood gas analysis with oxygen supplementation were also reviewed (pH, PvCO_2_, HCO_3−_, base excess (BE), lactate (Lac), ionized calcium (iCa)). A heparinized syringe was used to conduct blood gas analysis using the Blood Gas Analyser Gem Premier 3500 (instrumentation laboratory, Lexington, MA, USA).

### 2.2. Statistical Analysis

Body temperature, heart rate, respiratory rate, the diagnoses of cardiomyopathies, and venous blood gas analysis parameters, including pH, PvCO_2_, HCO_3_^−^, BE, Lac, and iCa, were compared between the 12 h death and 12 h survival groups. PvO_2_ was not included since variable concentrations of oxygen could be supplemented during the blood gas analysis. Since blood pressure was not recorded in all cats, it was compared between the smaller number of 12 h death and 12 h survival groups separately. PvCO_2_ values were also compared between cases with and without vasopressors or pleural effusion. Pairwise comparisons were performed using the Mann–Whitney U test with Bonferroni correction. Receiver operating characteristic (ROC) curves were created, and cut-off points were calculated by optimizing the findings for specificity and sensitivity. Spearman correlation analyses were used to calculate the correlations between PvCO_2_ and heart rate, respiratory rate, and body temperature. Since blood pressure was not recorded in all cats, the correlation analysis was conducted in a smaller number of cats.

### 2.3. Ethics Statement

The owners of all participating cats gave consent for the future use of clinical data at their presentation. No animal was identifiable in this publication; therefore, additional informed consent for publication was not required.

## 3. Results

### 3.1. Cases

Eighty-three cats were diagnosed with CPE between April 2019 and March 2021. Among them, 20 cats diagnosed with ATE, 15 cats without survival data at 12 h, nine cats treated using mechanical ventilation, and three cats without blood gas examination data were excluded. Thus, 36 cats were included in the study, and their clinical data and CPE diagnosis were retrospectively reviewed. No cat with the original CPE diagnosis was judged to be incorrect. The 36 cats included mixed-breed (n = 20), Scottish fold (n = 6), American shorthair (n = 4), British shorthair (n = 2), Bombay (n = 1), Maine Coon (n = 1), Ragamuffin (n = 1), and Russian blue (n = 1). The median age of the 36 cats was 8 years (range, 0.6–20.5 years). Twenty-one male cats (19 castrated) and 15 female cats (12 neutered) were included in the study. Eight of the 36 cats were previously diagnosed with cardiomyopathy and treated by primary care veterinarians. None of the 36 cats were previously diagnosed with CPE before their presentation to the hospital and prescribed diuretic drugs.

The median body temperature, heart rate, respiratory rate, and body weight at presentation were 37.0 °C (range, 32.0–39.4 °C), 166/min (range, 109–227/min), 86/min (range, 40–240/min), and 4.3 kg (range, 2.3–8.3 kg), respectively. Systolic blood pressure was measured in 31 of 36 cats, and the median value was 122 mmHg (range, 78–195 mmHg). Among 31 cats whose blood pressure could be measured, four cats were considered to be hypotensive (<90 mmHg). Blood tests, ultrasonographic tests, and radiographic tests were performed with oxygen supplementation in 36, 36, and 35 cats, respectively. The results of venous blood gas analysis are shown in Table 1. The cardiomyopathy phenotype was diagnosed as HCM (non-obstructive, n = 22; obstructive, n = 6), RCM (n = 7), and dilated cardiomyopathy (n = 1) by a cardiologist using radiographic and echocardiographic data. Pleural effusion was noted in 26 cats, and pericardial effusion was noted in five cats. Thirty-five cats were treated in oxygen cages using furosemide, and the median dosage of furosemide for 12 h was 2.95 mg/kg (range, 1–4.9 mg/kg). One cat died during treatment without furosemide in an oxygen cage because of severe hypotension. The following drugs were additionally used within 12 h: bronchodilator (n = 28), low-molecular weight heparin (n = 28), maropitant (n = 22), antibiotics (n = 23), carperitide (n = 20), pimobendane (n = 18), dobutamine (n = 11), norepinephrine (n = 8), dopamine (n = 5), and vasopressin (n = 2). Thoracentesis was performed in eight cats whose pleural effusion was severe enough to compromise ventilation.

The proportion of cats that survived to discharge was 25/36 (69%). Euthanasia was not selected in any cats. Median hospitalization time for the surviving cats and for those that died was 35 h (range, 9–156 h) and 10 h (2–60 h), respectively. Since the survival rate at 12 h was 78% (28 of 36 cats), twenty-eight cats were categorized in the 12 h survival group and eight in the 12 h death group.

### 3.2. Comparison of Basic Data and Blood Gas Analyses Using Survival Data

Clinical data and blood gas analyses were compared between the 12 h death and 12 h survival groups (Table 2). The body temperature in the 12 h death group was significantly lower than that in the 12 h survival group (corrected *p*-value = 0.01). The PvCO_2_ in the 12 h death group was also significantly higher than that in the 12 h survival group (corrected *p*-value = 0.005). Blood pressure was measured in 26 of 28 cats in the 12 h survival group and five of eight cats in the 12 h death group. Two of 26 cats in the 12 h survival group and two of five cats in the 12 h death group were hypotensive. Systolic blood pressure was significantly higher in the 12 h survival group (*p*-value = 0.015). Vasoconstrictor agents, such as dopamine, norepinephrine, and vasopressin, were used in 12 of 36 cats. Vasoconstricting agents were used in six of eight cats in the 12 h death group and in six of 28 cats in the 12 h survival group (*p*-value = 0.009). There was no significant difference among the diagnosis of cardiomyopathy between the two groups (*p*-value = 0.09). Other data were not significantly different between the two groups of cats. Receiver operating characteristic (ROC) analysis was conducted using body temperature and PvCO_2_; the area under the ROC curve (AUC) was 0.862 (95% confidence interval: 0.728–0.995) for body temperature (Figure 1a) and 0.879 (95% confidence interval: 0.749–1) for PvCO_2_ (Figure 1b). When the cut-off value of body temperature to predict a cat’s death within 12 h was set at 34.4 °C, the sensitivity was 62.5% and specificity was 96.4%. When the cut-off value of PvCO_2_ was set at 52 mmHg, sensitivity was 87.5% and specificity was 82.1%.

### 3.3. Association between PvCO_2_ and Other Data

Since PvCO_2_ was newly identified as a negative prognostic factor of cats with CPE that were not treated with ventilatory support in this study, the associations between PvCO_2_ and heart rate, respiratory rate, and body temperature were assessed (Table 3). A negative correlation was observed between PvCO_2_ and heart rate and body temperature. A weak negative correlation was observed between the PvCO_2_ and respiratory rate. In 31 cats that had their blood pressure measured, a negative correlation was observed between PvCO_2_ and systolic blood pressure. No significant difference in PvCO_2_ was noted between cats with and without pleural effusion (*p*-value = 0.448). The PvCO_2_ was significantly higher in cats treated with vasoconstrictor agents than in those not treated with the agents (*p*-value <0.001).

## 4. Discussion

In the present study, short-term prognostic factors in cats diagnosed with acute onset of cardiomyopathy-induced CPE were compared between cats that survived beyond 12 h and those that did not. Hypercapnia and hypothermia were identified as negative prognostic factors. Hypercapnia, as determined via venous blood gas analysis, may be an important prognostic factor for cats with CPE.

Hypothermia was an adverse prognostic factor in cases of cardiomyopathy-induced CPE in this study, and body temperature less than 34.4 °C was suggested to be a negative prognostic factor for death within 12 h. Hypothermia has also been reported to be a negative prognostic factor in a previous study on feline CHF [4]. Cats that died within 12 h were classified as showing moderate-to-profound hypothermia [15], which was reported to cause vasodilation and hypotension due to the reduced affinity of the α1-receptor for norepinephrine in rabbit [16]. Secondary hypothermia has been reported as a consequence of reduced heat production in critically ill patients, including those with cardiovascular diseases [17]. Hypothermia is also a negative prognostic factor in human patients with CHF [18]. Since body temperature can be measured frequently, it might be a useful prognostic factor not only at presentation but also during hospitalization.

A higher value of PvCO_2_, that is, hypercapnia, was found to be a negative prognostic factor for CPE in cats in this study. A cut-off PvCO_2_ value of 52 mmHg might be able to predict cat death within 12 h. In a previous study, HCM cats with CHF showed significantly higher total venous CO_2_ in comparison with those diagnosed with ATE, syncope, or subclinical HCM [19]. In that study, metabolic alkalosis caused by the administration of furosemide was considered a potential mechanism of hypercapnia in CHF. However, since no cat had been prescribed diuretic drugs before the onset of CPE, some mechanisms other than the administration of diuretic drugs were considered in the present study. Hypercapnia in arterial blood gas analysis has been reported to be a negative prognostic factor in human patients diagnosed with CPE [20,21]. In human patients, hypercapnia is considered to be a result of muscle weakness, diaphragmatic fatigue, reduced tidal volume due to obesity, or airway obstruction by alveolar edema fluid [20]. As in human patients diagnosed with CPE, diaphragmatic fatigue or airway obstruction by alveolar edema fluid might be more severe in the cats that died within 12 h than in those that survived beyond 12 h and might have caused significant hypercapnia. Meanwhile, hypotension was considered as another cause of hypercapnia in venous blood gas analysis [22]. Since hypotension and the use of vasoconstrictor agents was significantly associated with the death within 12 h and PvCO_2_, careful interpretation of PvCO_2_ is necessary especially in hypotensive cats.

Negative correlations between PvCO_2_ and body temperature, between PvCO_2_ and heart rate, and between PvCO_2_ and systolic blood pressure were found. CHF is the situation in which there is decreased cardiac output and results in hypotension, hypothermia, and bradycardia [4]. Since hypercapnia could be caused from diaphragmatic fatigue, airway obstruction by alveolar edema fluid [20], and hypotension [22], hypercapnia was considered to be as a result of CPE and hypotension due to CHF.

In the present study, 69% of cats with cardiomyopathy-induced CPE survived to discharge; this proportion is lower than that reported by others in cats with CHF [4,23]. In a study of cats diagnosed with CHF at an emergency service, 42 of 55 (76%) cats survived until discharge [4]. In that study, 32 of 55 (58%) cats were diagnosed with CPE, and the other cats were diagnosed with pleural effusion. In another study of cats diagnosed with CHF at a referral hospital, 32 of 34 (94%) cats, including 28 cats diagnosed with CPE, survived to discharge [23]. While cats diagnosed with CHF only with pleural effusion are curable with thoracentesis, CPE needs more intensive care and hospitalization. Considering these differences, the survival rate could be worse in the present study compared with previous studies. As other possibilities of the lower survival rate, the disease severity in the cats that presented to the hospital may have been higher than that in previous studies since the hospital only admitted emergency cases. In fact, four of 31 cats, who had their blood pressure measured at presentation, showed hypotension, and 12 of 36 included cats were treated using vasoconstricting agents. Since hypotensive CHF cats are generally in severe conditions, the included cats might have been in more severe conditions. These differences in the facilities or features of the included cases may have affected the survival rates.

There was no significant difference between 12 h survival and the diagnoses of cardiomyopathy. Since we included cats with CPE and excluded cats with ATE or pleural effusion, this result should be carefully interpreted. Long-term outcomes of feline CPE between different cardiomyopathies should be prospectively compared.

Ionized calcium was lower in the 12 h death group compared with the 12 h survival group with a nominal *p*-value. Decreased iCa is frequently reported in critically ill cats and is considered to be the consequence of vitamin D deficiency or resistance, acquired or relative hypoparathyroidism, or hypomagnesemia [24]. Calcium has a pivotal role in the sequence of myocardial excitation–contraction coupling and myocardial relaxation. Indeed, reversible CHF has been reported in a severe hypo-calcemic human patient [25]. The correlation of lower iCa and poor prognosis in cats with CPE and the pathophysiological contribution of iCa should be investigated in the future.

In conclusion, hypothermia and hypercapnia were identified as negative prognostic factors in cats diagnosed with cardiomyopathy-induced CPE. Venous blood gas analysis may be a new prognostic tool in feline cases of pulmonary edema, and a PvCO_2_ ≥52 mmHg may be used to discriminate more severe cases. Careful interpretation of the present study is needed since we included cats with cardiomyopathy-induced CPE and did not include cats diagnosed with ATE, treated by mechanical ventilation, or without venous blood gas analysis or survival data at 12 h.

This study had some limitations. Due to the retrospective nature of the study, the cases did not involve a uniform treatment regimen. A prospective case–control study should be conducted to minimize the influence of differences in treatment procedures. Since only cats diagnosed with CPE were collected, some cats without the statement of CPE as a diagnosis might be missed from this search. Moreover, since the number of included cases in the present study was relatively small, a larger number of cases should be prospectively included in future studies. Blood pressure measurement and perfusion evaluation are required to evaluate the relationship between hypotension and hypercapnia. However, these evaluations could not be conducted in all included cases because of their clinical instability. Since we used a non-invasive oscillometric method to measure blood pressure, it was difficult to obtain precise values, especially in hypotensive patients. Venous blood gas analysis was not conducted in three cats within 1 h after their presentation in this study. The clinical instability of cats could influenced the decision in terms of venous blood gas analysis. However, since the three cats excluded because of the absence of venous blood gas analysis within one hour were discharged from the hospital (data not shown), they did not seem to be in a severe condition compared with the included cases. In the present study, prognostic factors of feline CPE at more than 12 h could not be evaluated. Since feline CPE occurs and progresses rapidly, the short-term outcome is important in emergency facilities. However, longer term outcomes should be prospectively examined to reevaluate prognostic factors. In the present study, the diagnosis of cardiomyopathy was conducted using echocardiographic data obtained by veterinarians working at emergency rescue not a cardiologist. Although the diagnosis was made by a cardiologist, the preserved ultrasonographic data might not be adequate to diagnose. Ultrasonographic data should be obtained and diagnosed by cardiologists in a future prospective study.

## Figures and Tables

**Figure 1 vetsci-10-00232-f001:**
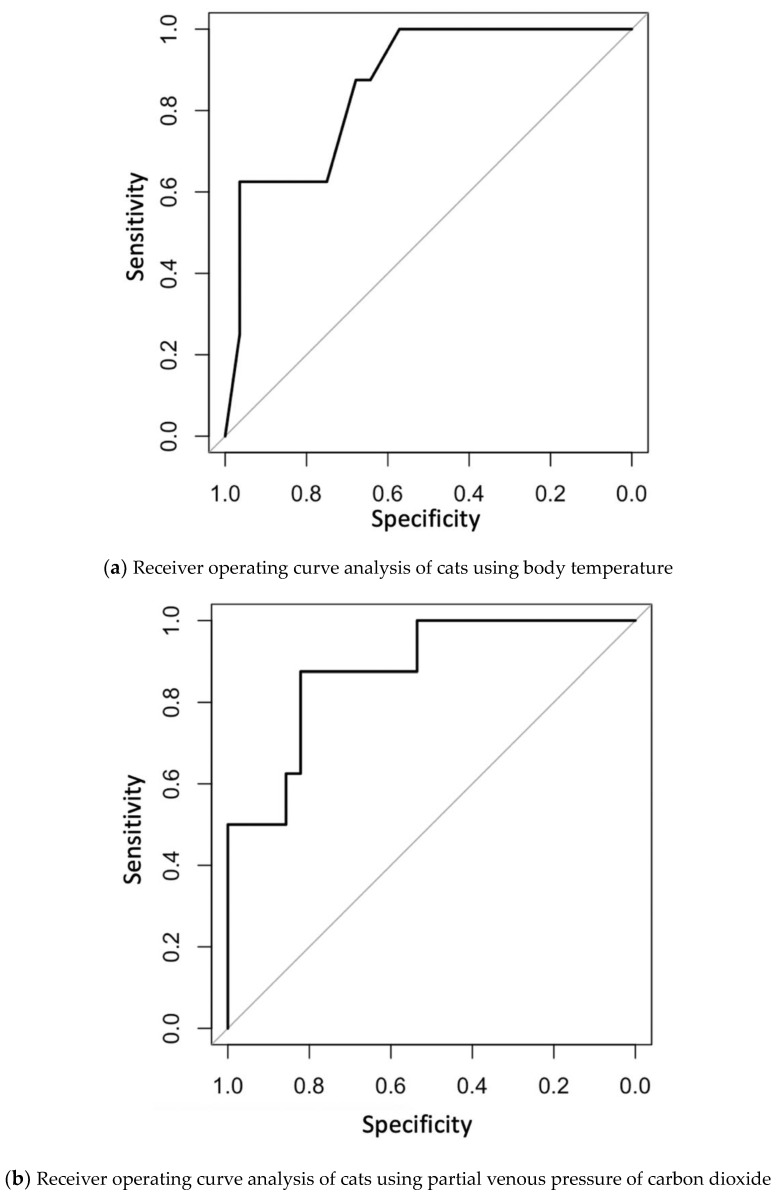
(**a**) Receiver operating curve analysis of cats to predict their death within 12 h using body temperature. (**b**) Receiver operating curve analysis of cats to predict their death within 12 h using partial venous pressure of carbon dioxide.

**Table 1 vetsci-10-00232-t001:** Blood gas analysis of cats diagnosed with cardiogenic pulmonary edema.

Variables	Median	Range (Minimum–Maximum)	Reference Value
pH	7.28	(7.02–7.41)	7.27–7.34
PvCO_2_ (mmHg)	48	(27–99)	37.1–44.5
HCO_3_^−^ (mmol/L)	21.1	(11.9–32.2)	19–21.2
BE * (mmol/L)	−6	(−17.1–5.8)	−6.7–−4.5
Lac * (mmol/L)	2.5	(0.7–11.2)	0.5–2.0
iCa * (mmol/L)	1.1	(0.72–1.34)	1.15–1.33

* BE, base excess; Lac, lactate; iCa, ionized calcium.

**Table 2 vetsci-10-00232-t002:** Comparison of clinical data and blood gas analysis of cats diagnosed with cardiogenic pulmonary edema.

Variables	Median (Minimum–Maximum) of 12 h Survival	Median (Minimum–Maximum) of 12 h Death	*p*-Value	Corrected *p*-Value ^#^
Body temperature (°C)	38.2 (32.0–39.4)	33.7 (32.0–37.0)	0.001	0.01
Heart rate (/min)	195 (105–227)	152 (132–211)	0.19	1
Respiratory rate (/min)	90 (54–240)	62 (40–150)	0.057	0.512
pH	7.30 (7.11–7.41)	7.17 (7.02–7.37)	0.033	0.3
PvCO_2_ (mmHg)	38 (27–70)	67 (46–99)	<0.001	0.005
HCO_3_^−^ (mmol/L)	18.7 (11.9–30.8)	24.8 (19.6–32.2)	0.097	0.869
BE * (mmol/L)	−7.7 (−17.1–5.8)	−3.4 (−11.2–2.9)	0.634	1
Lac * (mmol/L)	2.0 (0.7–9.1)	4.6 (1.1–11.2)	0.315	1
iCa * (mmol/L)	1.16 (0.8–1.34)	1.02 (0.72–1.23)	0.027	0.243

* BE; base excess, Lac; lactate, iCa; ionized calcium. ^#^
*p*-value was corrected using Bonferroni method.

**Table 3 vetsci-10-00232-t003:** Correlation between partial venous pressure of carbon dioxide and basic data.

Variables	Correlation Coefficient	*p*-Value
Body temperature	−0.649	<0.001
Heart rate	−0.499	0.002
Respiratory rate	−0.356	0.033
Systolic blood pressure	−0.431	0.016

## Data Availability

All data are shown in the manuscript.

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
