# Peer review of "The Prognostic Utility of Venous Blood Gas Analyses at Presentation in Cats with Cardiogenic Pulmonary Edema"

_vetsci, 2023, doi:10.3390/vetsci10030232_

Round 1
Reviewer 1 Report
In the study entitled “The prognostic utility of venous blood gas analyses at presentation in cats with cardiogenic pulmonary edema” the Authors aim was to investigate the association of physical examination and venous blood gas parameters with survival of cats with cardiogenic pulmonary edema (CPE).
General comments
This study improve the knowledge on the field. The subject of the work is of interest and that the topic of the manuscript falls within the journal topic. Authors rationale is worthy of investigation. However, the manuscript needs to be improved before it can be considered suitable for publication. Moreover, I think that the manuscript could be more suitable for case report category and not for article category.
Specific Comments
I suggest to avoid the use of personal form (i.e. our, we…) throughout the text.
Throughout the text several sentences are redundant. Please check and delete the repetition.
The title well reflects the main aim and findings of the work.
The abstract adequately summarize results and significance of the study. However, Authors should indicate the statistical analysis applied in the study.
The introduction section is well written and it falls within the topic of the study, and Authors cited appropriately bibliographic information.
The section of Materials and Methods is clear for the reader and it meticulously describes the methods applied in the study.
Regarding statistical analysis, Did Authors perform a normality test in order to test the normal distribution of data?
Results section as well as Discussion section is clear and well written, the findings obtained in the study were well discussed and justified with appropriate references.
The conclusion is well written, indeed Authors well summarize the results and the significance of the study. I suggest to move the last sentence relating to conclusion “In conclusion, hypothermia and hypercapnia were identified as negative prognostic factors in cats diagnosed with cardiomyopathy-induced CPE. Venous blood gas analysis may be a new prognostic tool in feline cases of pulmonary edema, and a PvCO2 ≥ 52 mmHg may be used to discriminate more severe cases.” before the limitation paragraph.
I suggest to improve the quality of tables and figures.
Authors should check and standardize the references in the list according to journal guidelines.
Reviewer 2 Report
Thank you, authors, for your hard work on the manuscript and for describing an interesting topic. Overall, I do have some concerns about the retrospective nature of the data collection and whether this information is generalizable going forward. I would recommend reworking the focus of the introduction, clarifying some major portions in your methods (particularly inclusion criteria), and further discussing some significant limitations. The paper also needs some English editing – I would encourage you to have complete editing of the paper performed by a native English speaker. I performed some grammar/syntax editing for the first paragraph or two of the paper but I have largely not marked issues throughout the rest of the paper. My specific comments and questions are below. Thank you again for your efforts on this paper.
Line 12: “ultimately” rather than “finally” might be a more appropriate word
Line 13: Change to “died” rather than “were died”
Line 14-15: This sentence is a bit repetitive given the prior sentence.
Lines 15-17: Awkward phrasing, sentence needs rewriting. E.g. Vasoconstrictor use was related to death…
Lines 20-21: Apologies, I could get access to the full article, but to double check, is CPE or CHF the leading cause of emergency hospital visits in this article?
Line 23: “develop” or “progress to” rather than “show”? Or even better, potentially get rid of the second portion of this sentence, as you restate this in your next sentence.
Lines 24-25: You’ve got to be careful of the phrasing here. You are implying in this sentence that ATE is a part of or is a sign of CHF, when those don’t necessarily have to go hand in hand. ATE is frequently seen concurrently with CHF but can also occur independently of CHF. Please rephrase to clarify. Your introduction of ATE here also kind of implies to your reader that this is an important aspect of your paper, whereas you actually exclude ATE cases from your study. So I think you also need to clarify your study population in this intro.
Line 26: This sentence needs rewriting – “significantly low survival period” is not quite right.
Line 32-35: Venous CO2 is not an accurate marker for ventilation in states of shock/hemodynamic compromise. This certainly needs to be addressed in your discussion but could be considered here as well.
Lines 36-37: I’m going to assume this PvCO2 was taken from unaffected limbs of ATE cats, but this is important to clarify.
Lines 25-26: Perhaps “has been assessed” would better reflect that venous blood gas analysis is used in many conditions but only evaluated in a few?
Line 42: The line “to save patients poorly responded” needs rephrasing.
Lines 41-49: I think this paragraph is tangential to your main point. You’re not assessing the need for mechanical ventilation here, so that literature is a little off topic. The flow from evaluating arterial pH and PCO2 in patients that might require intubation or ventilation to assessing venous blood gases in patients not treated with ventilatory support is not quite logical. I would rework this paragraph, reduce/eliminate discussion of mechanical ventilation here, and focus on the body of literature assessing CO2/acid-base abnormalities in general in dogs and cats and as prognostic markers. Help your readers understand – why are you looking at venous blood gases in this patient population? Is there a physiologic/pathophysiologic reason? Are there insufficient prognostic markers in this disease process? Are you simply looking at prognostic markers in different disease processes and using venous blood gas because its use is increasing?
Lines 52-53: You seem to narrow it down to a diagnosis of CPE later, so what were your search inclusion criteria? And is there a particular reason for this time frame? And is your aim to truly select cats with CPE or are you actually trying to diagnose cats with congestive heart failure? Obviously most of those cats with CHF will have CPE but pleural effusion alone is also described, so are you excluding cats with pleural effusion due to CHF but no CPE (and if so, why)?
Lines 59-60: Is there a reason you chose 12 hours as your survival outcome as opposed to 24 hours or survival to discharge? Would survival to discharge be more clinically relevant?
Line 60-61: I’m confused by this statement. So is euthanasia an exclusion criteria? If so, should it be listed with the others?
Lines 62-63: Do all cats get these if in respiratory distress? How many owners do this? What qualifies as a “blood test” (blood gas? CBC? chem?) and “ultrasonographic test” (like is that point of care? echo? abdominal ultrasound?)? It might be clearer to say what data/tests were collected if performed. It’s particularly important to specify when/why cats got venous blood gases, as there could be differences in populations that did and didn’t get venous blood gases. For instance, it’s easy to imagine that cats that were very unstable might not have gotten venous blood gases – this at a minimum needs to be addressed in your limitations.
Lines 66-68: Is this everything that comes on your panel or did you select these values for a reason?
Lines 72-73: I’m generally confused by your search and inclusion criteria. I imagine all medical records in your system were not reviewed, which is what this sort of implies. Please clarify search criteria that got you to the medical records that you reviewed, then inclusion criteria for your study. I’m also confused by the diagnosis of CPE. Lines 74-75 imply that radiographs alone were used to diagnose CPE, but then you go on in the rest of the paragraph to describe point of care echo (was that used?) and other imaging findings. I imagine it was a comprehensive evaluation by a cardiologist of clinical findings (please list), radiographic findings, and echo findings, but please clarify. You also later go on to describe different types of cardiomyopathies – what were the criteria you used to diagnose these? Someone should be able to replicate your classifications exactly if presented with your data. This also helps inform your limitations – for instance, if you only included cats that had both radiographs and echo, then you are leaving out a population.
Lines 75-76: You imply above that all cats had respiratory distress/dyspnea. Do some or all? Is that an inclusion criteria or no?
Line 87: How was normality assessed?
Lines 90-91: Was this pre-planned or decided on after this value came up as significant?
Lines 94-95: As this is a retrospective study, you would not have known you were collecting this data at the time of presentation. Is this performed for all animals in your hospital (as in all owners consent to release of clinical data for any possible future studies)? Or did you retrospectively call owners?
Lines 99-102: Again this paragraph is confusing in terms of your search/inclusion criteria. What does their “CPE diagnosis were reviewed” mean? Were some of the original CPE diagnoses judged to be incorrect? Could there be cats in respiratory distress without a “diagnosis” of CPE that you missed in your search? More clarification is needed.
Lines 113-114: This reads like a prospective and not a retrospective study. I imagine this was not truly standardized across your patient population. How many of your patients had each of these tests? If all of them did, please state so (and if they required this to be included in the study, please list these as inclusion criteria).
Line 121: For 12 hours or within the first 12 hours?
Lines 123-124: This is a high percentage of cats with CHF getting vasopressors – it is potentially worthwhile addressing this in a discussion point, as this is not a typical population.
Table 1: Please check reference value for BE.
Line 136: Basic Bata? Mean to be basic data I presume? And what do you mean by basic data – clinical parameters?
Figure 1: Not sure if this is a journal preference, but it would be nice to have a title on your figures so one could look at them and know the value being evaluated.
Lines 167-170: You did not state in your methods that you were going to compare these parameters.
Discussion: I would arrange your discussion such that you address your primary outcome in this paper first (after your first summarizing paragraph).
Line 180-182: You repeat yourself in these two sentences.
Lines 180-190: Reflecting points made earlier, this paragraph is a bit off topic from some of your main outcomes. For instance, you are citing literature that looks at survival to discharge whereas your main comparison groups were survival to 12 hours – why the difference? I think survival to discharge is arguably more relevant, but you should be comparing literature similar to your outcome. Additionally, you discuss literature about CHF, whereas your population is just CPE. You mention you excluded cases here of CHF without CPE but why? That’s not clear anywhere in your paper. Why is it important to you to exclude CHF without CPE, and if there is not a reason, why not look at a broader population? It’s tough to compare survival rates when you are talking about different populations. Overall, we need greater clarity and understanding from intro through to discussion about why you are choosing to evaluate this specific population.
Line 188: Can you think of any other reasons your patient population would be different? I would argue that any cat in congestive heart failure requiring oxygen supplementation would qualify as an emergency case.
Lines 196-197: Was reference 14 here meant to be reference 4? It might also be nice to qualify that reference 15 is a rabbit model.
Line 203: I would significantly qualify this statement, as your sensitivity and specificity for the predication of death within 12 hours is not perfect. There is a good deal of overlap in the ranges of CO2 in your two groups, so people should not come away with the thought that a CO2 of 52 is a death sentence for a cat.
Lines 204-205: Awkward phrasing of this sentence, please rephrase. Also was it arterial of venous CO2 that was evaluated in other studies, as this can make a significant difference.
Lines 209-217: You need to expand on and clarify these differentials for your readers. For one, the human populations you are comparing to are different from your population and are receiving noninvasive pressure support ventilation and CPAP, not conventional oxygen therapy like your patients. And while the differentials for arterial hypercapnia are absolutely reasonable to discuss, you have the major limitation of venous CO2 samples in many animals that appear from your description to be cardiovascularly unstable. Venous PCO2 is not representative of arterial PCO2 in hemodynamically compromised animals (the gap will be much greater than normal between the two), and you have a large population on vasopressors, which is not a normal CHF population. For all we know, these arterial samples could be 5-10mmHg lower (or more!), so it is very difficult to know what to do with this in this population. While it is still potentially reasonable to discuss as a prognostic indicator, PvCO2 in cardiovascularly compromised animals is not a pure reflection of ventilation (you start to address this with your comments on hypotension but I think further clarification is needed for your readership). This is a major limitation to the interpretation of venous CO2 values (at least to the underlying reason for this elevation) and should be explained carefully to your readers.
Lines 218-224: I think this paragraph strays from your main purpose and is questionably necessary. If you really want to discuss mechanical ventilation, typical cutoffs for concern for ventilation are PaCO2>60mmHg. Realistically people are rarely worried about a PvCO2 of 52mmHg in a critically ill cat (especially one that may be hemodynamically compromised), so we shouldn’t be suggesting that people should be thinking about mechanical ventilation in this range.
Lines 226-232: Your limitations need expanding. For one, I would say another limitation is the exclusion of euthanized cases. Obviously in veterinary medicine there are limitations with including euthanized cases as well, but either way introduces the potential for bias. What if euthanized cases were sicker (which they might well be)? Does this change your results? One way to test assess this bias is to see if your results change if you included euthanized cases, which I would strongly encourage you to do. Another major limitation is that you cannot actually say that hypercapnia was a negative prognostic factor in cats with CPE – you can only say that about the population of cats who got venous blood gases and whatever other inclusion criteria were needed, so that needs to be stated very clearly. I’ve mentioned some other limitations in above comments as well.
Lines 231-232: Are you stating that perfusion could not be assessed in these patients because of instability? Surely if a venous blood gas could be obtained, these patients had some perfusion parameters assessed?
Reviewer 3 Report
The study is well conducted and methodologically uncritical but it provides only a very small amount of information, information that everyone suspected since that is the reason why the blood gas measurement is carried out daily: hypercapnia predict mortality in dyspneic cats.
The scientific contribution is low.
This article should be submitted to a journal with a lower impact factor.
Inclusion criteria : was it cats that presented to the emergency for an acute cardiogenic pulmonary edema or was it a collection of data of all cats that were diagnosed with pulmonary edema ?
In the introduction, write a second objective : the association between PvCO2 and vital parameters (heart rate, respiratory rate and body temperature) since it was demonstrated in the article.
I'm not sure that the authors can talk about a cutoff value of 52 for the PvCO2 in the death group, since in the health group the values were between 27 and 70 mmHg.
In the discussion, the correlation between PvCO2 and other data (respiratory rate, heart rate, and body temperature) was not discussed.
